# A multi-ancestry GWAS of Fuchs corneal dystrophy highlights the contributions of laminins, collagen, and endothelial cell regulation

Bryan R. Gorman [1,2,31], Michael Francis [1,2,31], Cari L. Nealon[3], Christopher W. Halladay[4], Nalvi Duro[1,2], Kyriacos Markianos [1], Giulio Genovese [5,6,7], Pirro G. Hysi [8,9,10], Hélène Choquet [11], Natalie A. Afshari[12], Yi-Ju Li [13], VA Million Veteran Program*, J. Michael Gaziano[14,15], Adriana M. Hung[16,17,18], Wen-Chih Wu [19], Paul B. Greenberg[20,21], Saiju Pyarajan [1], Jonathan H. Lass[22], Neal S. Peachey [23,24,25,32] ✉ & Sudha K. Iyengar [23,26,27,32] ✉

Fuchs endothelial corneal dystrophy (FECD) is a leading indication for corneal transplantation, but its molecular etiology remains poorly understood. We performed genome-wide association studies (GWAS) of FECD in the Million Veteran Program followed by multi-ancestry meta-analysis with the previous largest FECD GWAS, for a total of 3970 cases and 333,794 controls. We confirm the previous four loci, and identify eight novel loci: *SSBP3*, *THSD7A*, *LAMB1*, *PIDD1*, *RORA*, *HS3ST3B1*, *LAMA5*, and *COL18A1*. We further confirm the *TCF4* locus in GWAS for admixed African and Hispanic/Latino ancestries and show an enrichment of European-ancestry haplotypes at *TCF4* in FECD cases. Among the novel associations are low frequency missense variants in laminin genes *LAMA5* and *LAMB1* which, together with previously reported *LAMC1*, form laminin-511 (LM511). AlphaFold 2 protein modeling, validated through homology, suggests that mutations at *LAMA5* and *LAMB1* may destabilize LM511 by altering inter-domain interactions or extracellular matrix binding. Finally, phenome-wide association scans and colocalization analyses suggest that the *TCF4* CTG18.1 trinucleotide repeat expansion leads to dysregulation of ion transport in the corneal endothelium and has pleiotropic effects on renal function.

Fuchs endothelial corneal dystrophy (FECD) is the most common corneal dystrophy, affecting more than 5% of people older than 40 years of age, and is the leading indication for corneal transplantation (keratoplasty) in the United States[1]. Globally, only one in 70 people needing a corneal transplant receive one[2], and a portion of transplants result in graft rejection or failure[3]. As surgical and pharmaceutical therapies are developed, genetically informed early diagnosis of FECD will be critical for directing treatment and preventing irreversible damage.

FECD is a progressive, bilateral disease[4]. Earliest indications of FECD are the presence of excreted collagenous deposits called guttae[4]. As FECD progresses, guttae grow in numbers and merge, leading to a thickening of Descemet's membrane. These changes put stress on corneal endothelial cells

(CECs), which regulate solute transfer and the flow of water into the stroma. CECs then begin to undergo cell death via apoptosis[5], accompanied by measurable changes in corneal biomechanics[6], CEC shape and density[7], and central corneal thickness (CCT)[8]. Disruption of endothelium function leads to corneal edema, resulting in blurred vision and, eventually, vision loss.

The etiology of FECD involves complex interactions of incompletely penetrant genetic factors with biological and environmental factors. Female sex and advanced age are established risk factors[4,9]. Risk may also differ across populations; lower rates of FECD diagnosis have been observed in African Americans in both clinical settings and Medicare claims[10]. Similarly, examining FECD by genetic ancestry in the Department of Veterans Affairs Million Veteran Program (MVP), we found significantly reduced

A full list of affiliations appears at the end of the paper. *A list of authors and their affiliations appears at the end of the paper. ✉e-mail: neal.peachey@va.gov; ski@case.edu

prevalence in participants of admixed African (AFR) and Hispanic/Latino (HIS) continental ancestries relative to European ancestry (EUR)[9].

The first genetic risk factors identified for FECD included ultra-rare mutations in *COL8A2* and *SLC4A11*[11]. Mutations in *COL8A2* cause the rarer early-onset form of FECD, which has a similar disease progression to late-onset FECD but is characterized by an abnormal distribution of collagen VIII[12]. Subsequently, genome-wide association studies (GWAS) identified four risk loci for FECD. Of these, the most significant is common variation at 18q21.2[12] tagging the CTG18.1 trinucleotide repeat (TNR) expansion in an intron of *TCF4* (transcription factor 4)[13]. As many as 75% of EUR FECD cases have at least one expanded CTG18.1 allele[11]. The previous largest FECD GWAS to date, Afshari et al.[14], confirmed *TCF4* and identified three additional loci: *LAMC1*, *KANK4*, and *ATP1B1*.

Recent genetic studies of FECD in non-EUR ancestries have largely focused on the genotyping and association of CTG18.1 alleles. CTG18.1 expansions are associated with FECD in African, Indian, Australian, and several East Asian populations[11]. However, CTG18.1 expansions are generally observed at lower frequencies in non-EUR FECD patients compared to EUR[10,15]. It remains unclear whether the population frequency of penetrant CTG18.1 alleles differs by genetic ancestry.

Here, we leverage genetic and clinical data provided by the MVP to conduct the largest GWAS analysis of FECD, and to the best of our knowledge, the first multi-ancestry meta-analysis. We confirm the four previously reported loci, including the presence of the *TCF4* locus in AFR and HIS, and present eight novel loci, expanding our knowledge of the genetic drivers of FECD.

## Results
### Multi-ancestry GWAS for FECD
We identified FECD cases in MVP participants of EUR, AFR, and HIS ancestry (Supplementary Data 1) following a clinically validated phenotyping algorithm[9]. Cases were mostly male (88.6%), reflecting the predominantly male composition of the MVP dataset[16]. As FECD is more common in women, there were more female cases than controls in each ancestry (combined 11.4% cases vs. 8.4% controls). Mean age of FECD cases ranged from 62.8 in AFR to 70.5 years in EUR. We performed a mixed-model GWAS for FECD in each ancestry (Fig. 1; Supplementary Fig. 1), including age, age-squared, sex, and ten ancestry-specific principal components as covariates.

The *TCF4* locus reached genome-wide significance (GWS; $P < 5 \times 10^{-8}$) across all three ancestries analyzed in MVP; to the best of our knowledge, this was the first time *TCF4* has been significantly associated with FECD in a GWAS in AFR or HIS (Table 1; Supplementary Fig. 1). The lead SNP at *TCF4*, rs11659764 ($r^2 = 0.21$ and $D' = 0.97$ with the previously reported FECD index variant, rs613872), was the same across all three ancestries. Although the marker varied in frequency, the additive effect of each allele of rs11659764 on FECD was highly similar across ancestries.

We applied local ancestry admixture mapping models at the *TCF4* locus in AFR and HIS to directly compare risk conferred by haplotype ancestry within the same individuals. In the AFR population, each EUR haplotype was additively associated with FECD (odds ratio (OR) = 1.28, 95% confidence interval = [1.02, 1.61]; $P = 0.015$), with 23% frequency of EUR haplotypes in cases vs. 18% in controls. In HIS, we found a similar OR for EUR haplotypes relative to AFR and Native American ancestry (NAT) haplotypes (OR = 1.27 [0.91, 1.78]; $P = 0.17$), with 64% EUR haplotype frequency in cases vs. 57% in controls, but this was non-significant due to lower power. Consistent with allele frequencies at our lead tagging SNP rs11659764, this result suggests that EUR haplotypes contain a higher frequency of pathogenic alleles compared to AFR and possibly also NAT haplotypes. The sample sizes for MVP Asian cohorts were too low to obtain reliable estimates[9], but data from prior studies in Japanese cohorts[17] and allele frequencies from the 1000 Genomes Project suggest that East Asians have lower FECD prevalence[4] due to lower frequency of CTG18.1 expansions.

**Fig. 1 | Study overview.** Genome-wide association study (GWAS) discovery analyses were performed in Million Veteran Program (MVP) European (EUR), admixed African (AFR), and Hispanic/Latino (HIS) cohorts. Numbers of Fuchs endothelial corneal dystrophy (FECD) cases and controls are shown. Afshari et al.[14] was included as a replication cohort. Follow-up analyses to interpret GWAS results are shown. HIS participants were not included in the multi-ancestry meta-analysis due to the low number of FECD cases. PGS, polygenic risk score; PheWAS, phenome-wide association study.

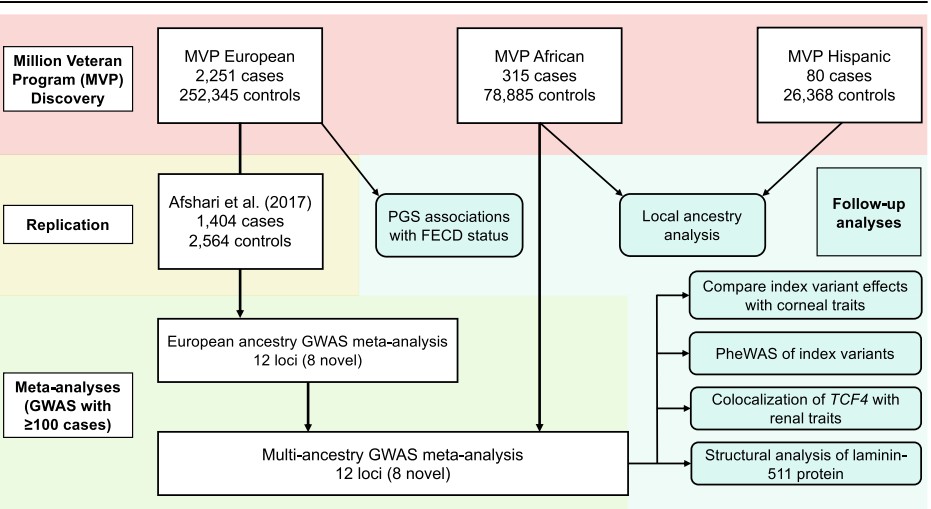

**Table 1 | Associations of the top single nucleotide polymorphism (SNP) at the *TCF4* locus, rs11659764, in Million Veteran Program (MVP) cohorts**

| Ancestry | Odds ratio [95% CI] | *P*-value | EAF cases | EAF controls |
|---|---|---|---|---|
| EUR | 6.41 [5.86, 7.01] | $9.4 \times 10^{-360}$ | 0.222 | 0.045 |
| AFR | 7.57 [4.87, 11.75] | $1.1 \times 10^{-19}$ | 0.061 | 0.009 |
| HIS | 7.16 [3.93, 13.04] | $6.2 \times 10^{-11}$ | 0.131 | 0.022 |

In European (EUR), admixed African (AFR), and Hispanic/Latino (HIS) cohorts, rs11659764 had the most significant association with Fuchs endothelial corneal dystrophy (FECD). The minor allele was associated with an increased odds ratio of FECD risk consistently across ancestry groups, despite differences in effect allele frequency (EAF). CI, confidence interval.

**Table 2 | Genome-wide significant loci in the multi-ancestry meta-analysis of Fuchs Endothelial Corneal Dystrophy (FECD)**

| rsID | Chr:Pos | Predicted causal gene | EA/ NEA | EAF | N case | N | OR [95% CI] | P-value | Direction |
|------|---------|----------------------|---------|-----|--------|---|-------------|---------|-----------|
| Novel loci ($P < 5 \times 10^{-8}$) | | | | | | | | | |
| rs11590557 | 1:54,324,099 | SSBP3 | A/G | 0.04 | 3655 | 258,564 | 1.61, [1.43 1.81] | $6.86 \times 10^{-15}$ | +?+ |
| rs74882680 | 7:11,700,254 | THSD7A | G/A | 0.02 | 3655 | 258,564 | 1.72 [1.48, 2.00] | $2.78 \times 10^{-12}$ | +?+ |
| rs150990106 | 7:107,955,927 | LAMB1 | A/G | 0.02 | 3655 | 258,564 | 1.75 [1.45, 2.10] | $4.33 \times 10^{-9}$ | +?+ |
| rs1138714 | 11:825,110 | PIDD1 | G/A | 0.53 | 3970 | 337,764 | 1.22 [1.16, 1.28] | $3.01 \times 10^{-14}$ | +++ |
| rs12439253 | 15:60,764,393 | RORA | T/G | 0.08 | 3970 | 337,764 | 1.29 [1.18, 1.40] | $4.31 \times 10^{-9}$ | +-+ |
| rs9303111 | 17:14,663,407 | HS3ST3B1 | C/A | 0.32 | 3970 | 337,764 | 0.81 [0.76, 0.85] | $1.17 \times 10^{-13}$ | --- |
| rs141208202 | 20:62,322,048 | LAMA5 | T/C | 0.05 | 3655 | 258,564 | 1.40 [1.25, 1.57] | $1.42 \times 10^{-8}$ | +?+ |
| rs114065856 | 21:45,432,844 | COL18A1 | T/C | 0.04 | 3970 | 337,764 | 0.61 [0.52, 0.72] | $2.87 \times 10^{-9}$ | --- |
| Previously reported loci | | | | | | | | | |
| rs79742895 | 1:62,317,189 | KANK4 | C/T | 0.04 | 3655 | 258,564 | 1.78 [1.59, 1.98] | $1.78 \times 10^{-24}$ | +?+ |
| rs1200114 | 1:169,091,251 | ATP1B1 | A/G | 0.66 | 3970 | 337,764 | 0.73 [0.69, 0.77] | $5.38 \times 10^{-34}$ | --- |
| rs2093985 | 1:183,125,187 | LAMC1 | T/C | 0.54 | 3970 | 337,764 | 0.80 [0.76, 0.84] | $2.58 \times 10^{-18}$ | --- |
| rs11659764 | 18:55,668,281 | TCF4 | A/T | 0.05 | 3655 | 258,564 | 7.15 [6.60, 7.74] | $8.60 \times 10^{-509}$ | +?+ |

Genomic risk loci from the meta-analysis of MVP European and African cohorts plus Afshari et al.[14]. We identified eight novel FECD loci and replicated all four previously reported loci. rs1138714 previously reached suggestive significance in Afshari et al. at $P = 7 \times 10^{-7}$. Genomic coordinates correspond to GRCh38. EA, effect allele; NEA, non-effect allele; EAF, effect allele frequency; OR [95% CI], odds ratio with lower and upper bounds of 95% confidence interval; Direction, SNP effect direction from MVP EUR, MVP AFR, and Afshari et al. meta-analysis cohorts, respectively; "?" indicates the AFR variant did not meet the allele frequency cutoff of 1% and was not included. Additional details can be found in Supplementary Data 3.

The MVP EUR discovery scan replicated all four known FECD GWAS loci[12,14] (*TCF4*, *KANK4*, *LAMC1*, and *ATP1B1*) and identified three novel loci at *SSBP3*, *THSD7A*, and *PIDD1* (Supplementary Data 2; Supplementary Fig. 1a). In Afshari et al.[14], a SNP at the *PIDD1* gene locus reached suggestive significance[14] ($P = 7 \times 10^{-7}$), and our lead novel variants at *SSBP3* and *THSD7A* were at least nominally significant ($P = 2.61 \times 10^{-5}$ and $P = 0.025$, respectively).

We then performed inverse variance-weighted fixed effects meta-analyses, first exclusively across the two European cohorts, MVP EUR and Afshari et al.[14] (Supplementary Fig. 2a). This EUR meta-analysis with 3655 FECD cases identified the four previously reported FECD loci as well as eight novel loci. Effect directions were the same for all twelve index variants in MVP EUR and Afshari cohorts (Supplementary Fig. 3a). Finally, we performed a multi-ancestry meta-analysis which added MVP AFR to the EUR-only meta-analysis (HIS were excluded due to fewer than 100 cases). This multi-ancestry meta-analysis tested a total of 18,302,074 variants in up to 3970 cases and 333,794 controls (Supplementary Data 1), ~2.8 times the case sample size of the previous largest FECD GWAS[14].

In the multi-ancestry meta-analysis, the four previously reported loci[14] attained GWS, and we identified the same eight novel FECD loci emerging at GWS from the EUR meta-analysis: *LAMA5*, *LAMB1*, *COL18A1*, *SSBP3*, *THSD7A*, *RORA*, *PIDD1*, and *HS3ST3B1* (Table 2; Supplementary Data 3; Fig. 2; Supplementary Figs. 4-15). Genomic control (λ) was 1.01, indicating minimal systematic inflation. Stepwise conditional and joint association analysis (COJO-slct) of the lead variant in each locus indicated no additional independent signals reaching GWS. (*TCF4* was excluded from conditional analysis due to the untyped CTG18.1 TNR expansion.)

As expected, the largest OR was observed at rs11659764 in *TCF4* (OR = 7.15 [6.60, 7.74]; Supplementary Fig. 13). Effect sizes at index SNPs were consistent across the MVP EUR and Afshari[14] cohorts, and all meta-analysis index SNPs were at least nominally significant ($P < 0.05$) in the prior GWAS, further validating our phenotyping approach. Six of twelve index variants did not meet the meta-analysis minor allele frequency (MAF) cutoff of ≥1% in AFR. Additionally, all index SNPs had consistent effect direction in AFR, with the exception of rs12439253 (*RORA*), which had a non-significant and opposite effect direction (Supplementary Fig. 3b). Two AFR SNPs, rs1138714 in *PIDD1* and rs114065856 in *COL18A1*, had consistent direction with the EUR cohorts but were not significant.

Linkage disequilibrium score regression (LDSC) analysis indicated the liability-scale SNP heritability (SNP-$h^2$) for FECD, based on EUR meta-analysis summary statistics, was 0.43 (standard error = 0.32), assuming a 5% population prevalence. As LDSC generally measures polygenicity[18], the uncertainty of the SNP heritability estimate may reflect the partially monogenic (*TCF4*) architecture of FECD.

## Novel FECD candidate genes

We identified candidate genes for our eight novel GWAS loci in the biological context of FECD; these are summarized in Table 3. Two novel loci emerged with lead variants in laminin genes: *LAMA5* (α5) and *LAMB1* (β1). Together with the previously reported *LAMC1* (γ1) protein, these subunits form the laminin-511 heterotrimer (LM511; also called laminin-10), implicating an important role for LM511 in CEC maintenance and FECD pathogenesis. In previous studies, LM511 staining patterns were thicker in FECD corneas than controls[19]; additionally, LM511 facilitated the expansion of CECs in culture[20] and promoted recovery of CECs in animal models of CEC transplantation[21]. At *LAMB1*, our association peak consisted of three low-frequency (1–2% in EUR) variants in LD ($r^2 > 0.9$; Supplementary Fig. 16), each with a posterior inclusion probability (PIP) of 30–35% estimated from SuSiE fine-mapping[22,23]. Of these, the most likely causal variant is the missense mutation at rs80095409 (p.Arg795Gly), which was computationally predicted to have a deleterious impact on protein structure by both SIFT[24] and Polyphen[25] classifiers, with a Combined Annotation Dependent Depletion (CADD) score of 29.7[26] (Supplementary Fig. 16). Interestingly, the *LAMB1* locus has no pleiotropy with other ocular traits reported in the GWAS Catalog (Supplementary Data 4).

At *LAMA5*, the characteristic subunit of LM511, the lead variant rs141208202 is also a low-frequency (4–5% in EUR) missense mutation, p.Gly2156Glu, that is predicted by SIFT to be deleterious and had 78% PIP estimated by SuSiE. The next most significant variant (rs143905087; $P = 6.74 \times 10^{-8}$), is an intronic variant in *CABLES2* in only moderate LD with the lead variant ($r^2 = 0.54$; Supplementary Fig. 17) and 17% PIP. Thus, we prioritize rs141208202 as a likely causal variant at *LAMA5*, mediated through putative impact on protein structure, which we explore further below. However, rs141208202 is a *LAMA5* splicing quantitative trait locus (sQTL) in some tissues in GTEx[27] and is located within a CTCF binding site[28], and thus may also have a regulatory impact.

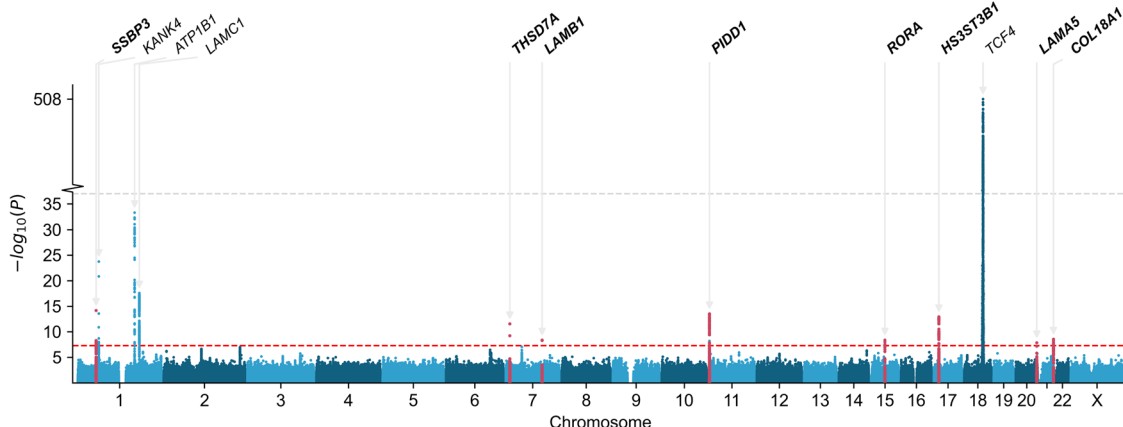

**Fig. 2 | Manhattan plot of the Fuchs endothelial corneal dystrophy multi-ancestry meta-analysis.** Plot shows the $-log_{10}(P)$ for associations of genetic variants with Fuchs endothelial corneal dystrophy across 22 autosomal chromosomes plus chromosome X. Genome-wide significant loci are labeled by names of candidate genes; novel loci are highlighted in red with bold gene names. The red line indicates the genome-wide significance threshold ($P < 5 \times 10^{-8}$). A y-axis break is used to include the most significant variant at *TCF4*.

## Table 3 | Summary of novel candidate genes

| Novel candidate gene | Putative function |
| --- | --- |
| *SSBP3* | Likely binds to polypyrimidine promoter of *COL1A2*, regulating transcription. |
| *THSD7A* | Regulator of endothelial cell migration and adhesion via binding of integrin αvβ3. |
| *LAMB1* | Beta-1 subunit of laminin-511; component of basal lamina. |
| *PIDD1* | May regulate corneal endothelial cell death via apoptosis. |
| *RORA* | Regulator of genes involved in circadian rhythm and oxidative stress. |
| *HS3ST3B1* | Regulator of heparan sulfate, which may have a role in corneal homeostasis. |
| *LAMA5* | Alpha-5 subunit of laminin-511; component of basal lamina. |
| *COL18A1* | Collagen type XVIII subunit; cleaved to form endothelial cell regulator endostatin. |

Candidate genes for eight novel loci identified in the multi-ancestry meta-analysis for Fuchs endothelial corneal dystrophy.

We discovered two novel loci likely driven by collagen genes: *SSBP3* and *COL18A1*. SSBP3 (single-stranded DNA binding protein 3) is predicted to bind to a polypyrimidine tract in the promoter of *COL1A2* and regulate its expression[29]. Collagen type I is one of the primary collagens found in corneal tissue, and other subunits of collagen type I have emerged in previous GWAS of corneal traits[30]. *COL18A1* encodes the alpha chain of type XVIII collagen, a ubiquitous component of the basement membrane (BM). In addition to its structural role, cleavage of type XVIII collagen generates the regulatory peptide endostatin, which inhibits proliferation of vascular endothelial cells through G1 arrest[31] and can induce cell death, implicating anti-tumorigenic and anti-angiogenic properties of this domain[32].

Another novel locus was identified at *THSD7A*. *THSD7A* interacts with integrin alpha V beta 3 (αvβ3)[33,34], expressed on CECs[35], to inhibit migration. *THSD7A* has been previously associated in GWAS studies with four ocular traits: glaucoma, intraocular pressure, refractive error, and cataract (Supplementary Data 4). Though the lead variants for these associations are in *THSD7A*, they have low $r^2$ with our lead SNP rs74882680 ($r^2 \leq 0.015$), due to the presence of multiple distinct LD blocks within this gene (Supplementary Fig. 8).

We identified a gene-dense region at 11p15.5 tagged by lead SNP rs1138714 that contained several potential candidate genes (Supplementary Fig. 10). We fine-mapped the EUR meta-analysis and found one credible set with 16 SNPs. The SNPs in the credible set with the highest PIP, as well as the highest CADD score, were located primarily within *PIDD1*, but also within and surrounding *PNPLA2* (Supplementary Fig. 18). *PIDD1* has a potential role in FECD by regulating CEC death via apoptosis. *PNPLA2* (also known as desnutrin or TTS-2.2) is a paralogue of *PNPLA4* (hGS2), which is responsible for transferring fatty acids from triglycerides to retinol, as well as hydrolyzing retinylesters[36]. Adequate retinol is required for corneal development and function, and CECs are involved in the conversion of retinol into retinoic acid[37]. *PNPLA2* and *PIDD1* were differentially expressed in CEC in patients with keratoconus (KC) and myopia[38]. Another biologically relevant nearby gene is *CD151*, a global regulator of endothelial cell-cell and cell-matrix adhesion[39]. *CD151* gene product is a member of the tetraspanin family and, along with type XVIII collagen and laminins, is a member of the collagen chain trimerization pathway.

The association of rs1138714 with eQTLs at all three of these biologically relevant genes in GTEx indicates that synchronized co-expression of multiple causal genes in this region may also be possible[27]. This locus has been previously associated with multiple ocular traits, and our FECD index variant at rs1138714 is in LD with rs10902223 ($r^2 = 0.99$), reported as the lead variant for KC and intraocular pressure, and is also in moderate LD with rs4963153 ($r^2 = 0.54$), the lead variant reported for associations with corneal resistance factor (CRF) and CCT (Supplementary Data 4).

We identified a novel association with FECD at *RORA*, which belongs to the family of retinoic acid-related orphan receptors (RORs). RORs are a superfamily of nuclear receptor transcription factors which bind to hormone response units. Although RORA shares structural features with retinoic acid receptors (RARs), it does not have known ligand-binding properties with retinol. *RORA* is commonly associated with regulation of *BMAL1* and circadian rhythm; CECs have a highly robust circadian clock, and FECD and other corneal maladies are known to exhibit diurnal variation[40]. *RORA* is induced by oxidative stress; reduction of *NFE2L2* nuclear factor translocation, which leads to downregulation of antioxidant

**Fig. 3 | Comparing effects of Fuchs endothelial corneal dystrophy (FECD) index variants with four corneal traits.** Variant-level comparison of FECD variants with four other corneal traits: keratoconus (KC), central corneal thickness (CCT), corneal resistance factor (CRF), and corneal hysteresis (CH). Box sizes correspond to *P* value tiers, and * indicates *P* < 0.05. Units: FECD, odds ratio; KC, odds ratio; CCT, μm; CRF, mm Hg; CH, mm Hg.

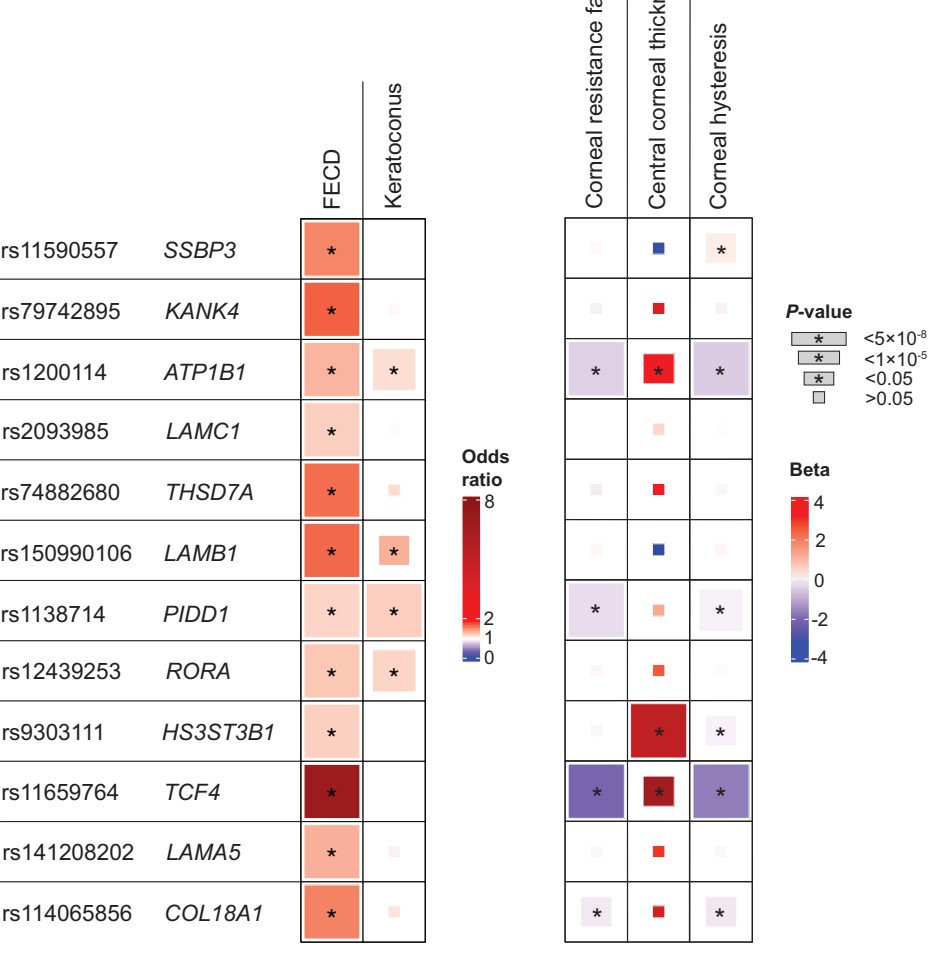

## Pleiotropy of FECD risk alleles

We compared the effect size and direction of our lead FECD variants with summary statistics from other corneal traits: KC[30], CCT[47–49], CRF[50,51], and corneal hysteresis (CH)[51,52]. We found consistent directional trends in a variant-level comparison across these traits (Supplementary Data 5). Eight of twelve FECD index variants had nominally significant associations (*P* < 0.05) in at least one other corneal trait. At the nominally significant variants for each respective trait, all KC and CCT variant effects were in the same direction as FECD, while all CRF variants, and all variants but one in CH (*SSBP3*) were associated with effects in the opposite direction (Fig. 3; Supplementary Data 6). The relationship of genetic effects of CRF and CH with those of FECD were directionally consistent with previous observational reports[53]. Genetic correlations ($r_g$) between FECD and other ocular traits were not significant, however they followed the same directional pattern as the variant-level trends.

We calculated polygenic scores (PGS) for every trait in the PGS Catalog[54], in all MVP EUR subjects. To discover shared genetic etiology with other traits, we performed a phenome-wide scan for the association of normalized PGS scores with FECD case-control (Supplementary Data 7). A total of 2,649 scores corresponding to 560 uniquely mapped traits in the Experimental Factor Ontology (EFO) were tested; we considered 24 traits to be significant after multiple testing correction (*P* < 0.05/560). We found that PGSs for other corneal traits had the strongest associations with FECD, including CH (OR = 0.83 [0.79, 0.86]; *P* = 7.04 × 10⁻²⁰) and CRF (OR = 0.86 [0.83, 0.90]; *P* = 4.73 × 10⁻¹²). The negative effect direction of these corneal trait PGS associations is consistent with our variant-level analysis in Fig. 3.

expression, has previously been observed in FECD cases[41]. RORA also regulates the differentiation and maintenance of type-2 innate lymphoid cells[42], which are among the immune cells resident in the cornea[43]. Additionally, our top FECD index SNP at *RORA*, rs12439253, has $r^2$ = 0.59 with the KC index SNP rs76194223.

Finally, a novel FECD locus was found in an intergenic region ~314 kb downstream from the nearest coding gene, *HS3ST3B1*. HS3ST3B1 is a 3-O-sulfotransferase integral membrane protein, which catalyzes the addition of sulfate groups to heparan sulfate (HS). HS is required for a wide range of cellular processes, including maintaining corneal homeostasis in CECs[44]. Heparanase, which acts as a protease of HS in the BM, was overexpressed in keratoconic corneas, and heparanase catalytic activity was correlated with KC severity[45]. In addition, a severe impediment to corneal wound healing was observed in a mouse HS knockout model[44]. This locus has been previously associated with CCT in three GWAS studies ($r^2$ = 0.95–1) and with CEC size variation coefficient ($r^2$ = 0.63) (Supplementary Data 4).

Intriguingly, a locus near *ANAPC1* previously reported to account for 24% of variability in CEC density in an Icelandic population[46] reached suggestive levels of significance in our multi-ancestry meta-analysis. However, the allele reported to decrease CEC density (rs78658973-A) was protective for FECD (OR = 0.86 [0.80, 0.92]; *P* = 5.1 × 10⁻⁶). In the same study, this allele was also significantly associated with increased coefficient of cell size variation and decreased percentage of hexagonal cells. The other allele reported to decrease CEC density, the CTG18.1 TNR expansion, greatly increases risk of FECD (Supplementary Data 4). Thus, our results support a complex relationship between CEC density and FECD.

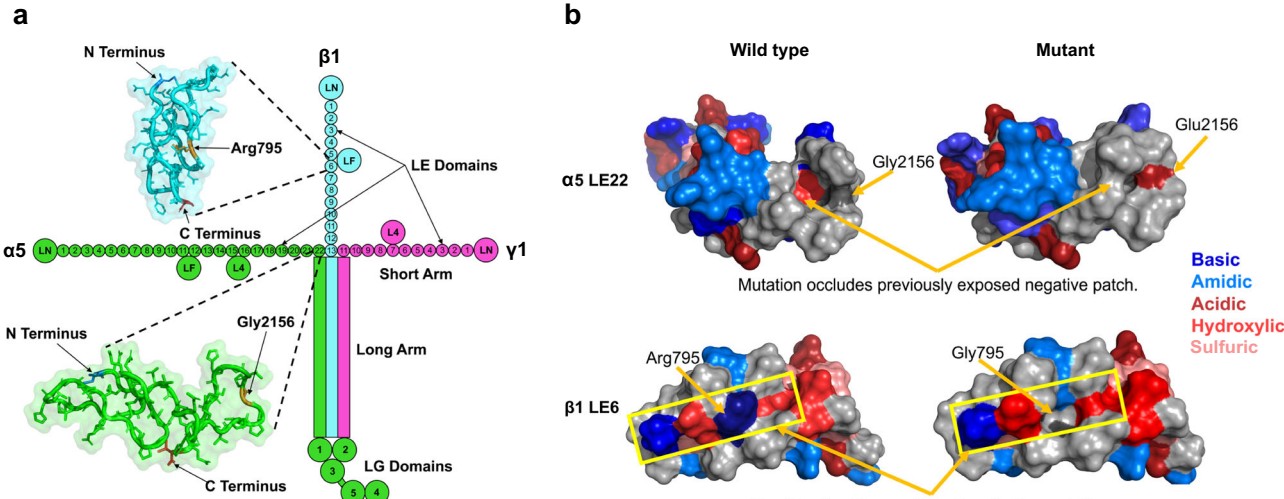

**Fig. 4 | Structure of LM511 and predicted impact of missense variants in laminin genes *LAMA5* (α5) and *LAMB1* (β1). a** Structural organization of the laminin-511 (LM511) heterotrimer. The green color denotes the LAMA5 subunit, blue denotes LAMB1, and pink denotes LAMC1. Significant FECD variants are located on the short arms of α5 and β1, in LE (laminin-type epidermal growth factor (EGF) like) domains LE22 and LE6, respectively. The insets depict AlphaFold 2 predictions of these domains, and the locations of the mutated residues are shown in orange. **b** (Top) Predicted surface structure of the α5 LE22 domain with and without the Gly2156Glu variant. (Bottom) Predicted surface structure of the β1 LE22 domain with and without the Arg795Gly variant.

After corneal traits, several renal PGSs had significant associations with FECD status, including urinary albumin-to-creatinine ratio (UACR; OR = 1.15 [1.10, 1.20]; $P = 2.28 \times 10^{-10}$), which was reported previously[55], plus urinary sodium (OR = 0.89 [0.86, 0.93]; $P = 8.80 \times 10^{-8}$) and urinary potassium (OR = 0.91 [0.87, 0.95]; $P = 6.46 \times 10^{-6}$).

We then performed phenome-wide association scans (PheWAS) using the index variants from the FECD meta-analysis (Supplementary Data 8). In up to 458,296 MVP EUR participants, a total of 1460 phenotypes were tested for each SNP: 1170 phecodes[56], 64 laboratory and vital signs measurements, and 225 survey questions. We found 32 associations with non-corneal traits that were significant after multiple testing correction ($P < 0.05/17,520$). Among the significant pleiotropic associations of FECD risk alleles are a protective association with open-angle glaucoma at *SSBP3*, risk-increasing associations with benign colon neoplasms at laminin genes *LAMA5* and *LAMC1*, and an association with increased heart rate at *LAMB1*, which is replicated in the UK Biobank ($P = 2.0 \times 10^{-8}$)[57].

The most significant PheWAS associations were observed at the *TCF4* risk allele (Supplementary Fig. 19), which was strongly associated with laboratory measurements of increased serum bicarbonate ($P = 7.0 \times 10^{-62}$), decreased chloride ($P = 9.1 \times 10^{-24}$), and increased potassium ($P = 2.3 \times 10^{-9}$), followed by decreased platelet ($P = 1.3 \times 10^{-7}$), monocyte ($P = 1.9 \times 10^{-7}$), and neutrophil ($P = 2.1 \times 10^{-6}$) counts. The pleiotropic association with serum bicarbonate likely explains the significant association with this trait we previously observed in a phenome-wide comorbidity scan of FECD case-control status[9].

Upon further evaluation of the *TCF4* locus in these significant laboratory measurement traits, we found that the index SNP of each trait (rs11659764) was the same as in FECD. Each trait displayed a highly similar complex pattern of local associations (Supplementary Fig. 20), which in FECD are thought to be caused by the partial LD of SNPs on different haplotypes with pathogenic CTG18.1 alleles. This same pattern was observed using externally derived UACR summary statistics[55], validating our results. We found that the regression coefficients of significant SNPs at the *TCF4* locus were highly correlated across FECD and each of the four laboratory-measured renal traits, suggesting colocalization. Positive correlation with FECD was observed in effect direction and magnitude for bicarbonate ($r = 0.91$), potassium ($r = 0.77$), and UACR ($r = 0.95$), while negative correlation was observed with chloride ($r = -0.90$).

To further untangle the pleiotropic effects with renal traits, we performed Bayesian colocalization analyses under the assumption of a single causal variant (the untyped CTG18.1 expansion) using coloc[58]. All four traits showed evidence of colocalization, with posterior probabilities >0.999 (Supplementary Data 9). We consider these findings to be strong evidence that the CTG18.1 expansion has pleiotropic effects on renal function. Moreover, the strength of the association with serum bicarbonate suggests that the effect of the CTG18.1 expansion on FECD may be mediated through dysregulation of ion transport in CECs.

## Structural analysis of two coding laminin variants

The gene products of two novel FECD loci at *LAMA5* and *LAMB1*, plus the known locus at *LAMC1* (which we replicated), are the three subunits of the LM511 heterotrimer. Each monomer (α5, β1, and γ1) of LM511 is a multidomain polypeptide; these interact with each other to form the long arm of a cross-shaped structure, while their non-interacting portions constitute three short arms (Fig. 4a). The short arms are composed of laminin-type EGF-like (LE) domain repeats that terminate in a laminin N-terminal (LN) domain[59]. These short arms interact with other extracellular proteins to assemble and stabilize the BM, while the long arms facilitate interaction with cell surface receptors via globular domains.

The missense mutations at rs141208202 (*LAMA5*) and rs150990106 (*LAMB1*) correspond to a glycine to glutamic acid substitution at position 2156 of α5 LE22 and an arginine to glycine substitution at position 795 of β1 LE6, respectively. We examined the potential impact of these mutations on the structure and function of LM511, using SWISS-MODEL[60] and AlphaFold 2 (AF2)[61] to model the α5 LE22 and β1 LE6 domains (Supplementary Fig. 21a, b).

The glycine to glutamic acid substitution in α5 LE22 replaces a small hydrophobic residue with a large acidic one, altering the surface hydrophobicity and topology (Fig. 4b, top). The required orientation of α5 LE22, with respect to the cross, positions the mutated residue in proximity to the other chains. This substantial change in LE22 may disrupt inter-chain interactions and could also potentially destabilize the triple-helix of the long arm, leading to disrupted interactions with cell surfaces through allosteric modulation of the LG domains.

Replacing the large basic arginine in β1 LE6 with a smaller hydrophobic glycine induces similar changes in surface hydrophobicity and topology (Fig. 4b, bottom). The wild-type arginine is part of a positive-negative-positive-negative patch on the LE6 domain surface that is likely to constitute a binding motif. Breaking this motif can disrupt interactions to neighboring β1 domains or to other extracellular matrix

proteins, resulting in binding affinity differences to the BM and altered cell signaling.

While these two mutations have high potential to disrupt inter-domain interactions, it is unlikely that they will induce significant changes to the tertiary structures of α5 LE22 and β1 LE6. This is because LE domain backbones are covalently linked through four disulfide bonds that prevent any significant deviations from the native fold (Supplementary Fig. 21c), resulting in no change in intra-hydrogen bond count for α5 LE22, and a loss of only three hydrogen bonds in β1 LE6 (Supplementary Fig. 21d). Correspondingly, Duet[62] predicted that the β1 LE6 mutation is more destabilizing than the α5 LE22 mutation. Overall, our structural analysis suggests that the variants associated with FECD may destabilize LM511 through altered inter-domain interactions, rather than through structural changes of the mutated domains.

## Discussion

In this study, we have identified eight novel genomic risk loci for FECD, and replicated the four existing loci, in the largest GWAS of FECD cases to date ($N_{cases}$ = 3970). Our multi-ancestry analysis confirmed the considerably large effect of the *TCF4* locus across AFR and HIS ancestries; *TCF4* was the exclusive signal reaching GWS in these ancestry groups with fewer cases. Our results increase confidence in known FECD mechanisms, and our novel candidate genes expand our understanding of the contributions of laminins, collagen, integrins, and CEC regulation in FECD pathophysiology.

All three genes encoding subunits of LM511 had GWS associations with FECD in this study. LM511 has been primarily studied in the context of tumor growth, both in vitro and in vivo, in relation to integrin-mediated adherence to tumor cells[63]. In a recent US cohort study where 68% of FECD cases were female, FECD was associated with higher risks of breast, thyroid, ovarian, and basal cell carcinomas[64]. Our PheWAS results also indicate the index variants at *LAMC1* and *LAMA5* are significantly associated with colon cancer (Supplementary Data 8). These findings suggest a potential link between LM511 and the increased risk of certain cancers observed in FECD cases. Additionally, our structural analysis of mutations in *LAMA5* and *LAMB1* suggests that disruption of LM511 inter-domain interactions or extracellular matrix binding increases risk of FECD.

Collagens are major components of the BM and Descemet's membrane, and the infiltration of collagenous secretion (guttae) from Descemet's membrane is a hallmark of FECD. Our GWAS results included novel associations with *COL18A1* (type XVIII collagen) and *SSBP3*, whose gene product putatively regulates *COL1A2* (type I collagen). Type XVIII collagen contains a laminin-G-like/thrombospondin-1 (LAM-G/TSP-1) homology region and thus exhibits structural similarity to laminins and thrombospondins such as THSD7A[32]. Additionally, Type XVIII collagen is an HS proteoglycan; the product of another novel gene *HS3ST3B1* is responsible for generating binding sites for proteins on HS chains. FECD CEC samples have been previously shown to contain higher levels of keratan sulfate, a sulfated glycosaminoglycan (GAG) found in the ECM[41], and our results suggest HS-GAGs may have a similar role in FECD related to lubrication.

Our findings further highlight the importance of dysregulated ion balance in FECD and indicate a pleiotropic connection to kidney function at *TCF4*. In addition to the known association between the UACR PGS and FECD[55], we found associations with PGSs for urinary sodium and urinary potassium, driven by shared signals at the *TCF4* locus. A PheWAS on our *TCF4* index variant (rs11659764) also revealed associations with serum measurements of bicarbonate, calcium, and potassium (Supplementary Fig. 19). Using GWAS summary statistics for these traits, we demonstrated a high probability of co-localization between associations for FECD, UACR, and serum ion measurements at *TCF4* (Supplementary Fig. 20). As highly associated FECD SNPs at *TCF4* are considered to tag CTG18.1 TNR expansion alleles, colocalization implies an underlying association of these with UACR and serum ion levels as well.

Similarities between ion transport in CECs and in the proximal tubule cells of the kidney, both forming leaky epithelia, have long been observed[65].

The convergence of evidence across FECD and serum and urinary ionic concentrations suggests that the pathogenicity of CTG18.1 expansions is mediated through dysregulated ion balance in CECs. This may be a consequence of modified gene expression, RNA toxicity, or other mechanisms. Notably, an analysis of corneal endothelium in samples of FECD with CTG18.1 expansions found increased expression of genes involved in ion transport[66]. Consistent with the corneal endothelium's role as a pump, ion transport is a major theme of FECD genetics, most famously in the association of highly penetrant rare mutations in solute transporter *SLC4A11*. Our analysis also replicated the GWAS locus at *ATP1B1*, whose gene product regulates sodium balance as a subunit of a $Na^+/K^+$ ATPase.

Our analysis contains several limitations. First, the algorithm we used to identify FECD cases[9], while clinically validated, was based solely on electronic health record diagnoses, and not the slit lamp imaging used previously[14], which may have diluted the phenotyping in our analysis. We were constrained by the demographics of FECD cases in the MVP dataset; FECD is more common in women, but our sample, and MVP in general, skew heavily male, which had the potential to bias our GWAS towards the identification of male-specific genetic factors. However, because our novel index variants were all at least nominally significant in ref. [14] (68% female FECD cases), with consistent effect estimates (heterogeneity $P > 0.05$; Supplementary Figs. 4–15), our results may indeed be generalizable to both males and females. We also did not differentiate between rare early-onset and more common late-onset FECD, whose pathophysiologies may involve separate genetic mechanisms[67].

It is well established that the most predictive FECD allele at 18q21.2 is the CTG18.1 TNR expansion[11]. Our GWAS used chip-based genotyping, so we relied on SNPs tagging CTG18.1 alleles instead of direct genotyping. Although our lead *TCF4* SNP rs11659764 is an imperfect proxy for CTG18.1, it nonetheless showed a strong and consistent association signal across multiple ancestry groups.

AlphaFold 2 and SWISS-MODEL are accurate in single-state predictions, but a limitation arises as they provide no information on protein fluctuations, leading to lower confidence in the structure of intrinsically disordered regions (IDR). While crystal structures of homologs suggest that there are no significant IDRs in LM511 LE domains, the question of how mutations can affect their dynamics remains. Although the predicted single-state structures used here do not capture shifts in dynamics, they nonetheless inform that the mutations significantly change surface chemistry and topology, and by extension, interactions to binding partners. Additionally, AlphaFold 2 is trained on wild-type protein structures and therefore has limited ability to predict when missense mutations will cause changes in protein folding[68]; however, the backbone structure of LM511 indicates that folding changes are not likely to occur from our FECD risk alleles, and so this limitation should not impact our functional predictions.

Our GWAS results have tripled the number of genomic risk loci associated with FECD, from four to twelve. We were able to place these novel loci into biological context compatible with currently understood mechanisms of FECD disease progression. Additionally, the MVP dataset enabled unprecedented quantitative analyses of non-EUR cohorts[16], and this analysis expands our understanding of the shared genetic architecture of FECD in these populations. We hope these results will lead to improved genetic risk prediction and, once experimentally validated, will help inform modern treatment strategies.

## Methods

### Ethics/study approval

The VA Central Institutional Review Board (IRB) approved the MVP024 study protocol. Informed consent was obtained from all participants, and all studies were performed with approval from the IRBs at participating centers.

### Phenotyping

We used a rules-based algorithm[9] based on structured electronic health record (EHR) data, specifically International Classification of Diseases

Clinical Modification and Current Procedural Terminology codes, the accuracy of which was confirmed at three VA Medical Center Eye Clinics[9]. Cases were identified based on the presence of FECD codes (371.57 for ICD-9-CM; H18.51 for ICD-10-CM) on two separate visits and the absence of ICD-9-CM or ICD-10-CM codes for confounding corneal conditions or complicated intraocular surgeries. Controls without FECD were identified as having undergone at least one eye exam, with no codes for FECD, confounding corneal conditions, or complicated intraocular surgeries. We applied this algorithm to conduct GWAS and to analyze associated EHR data.

## QC and imputation
MVP samples were genotyped on the ThermoFisher MVP 1.0 Axiom array. The design and QC of the array is described in detail elsewhere[69]. Genotypes were phased using SHAPEIT4[70] and imputed to the TOPMed reference panel (version r2) using Minimac4.

## GWAS
Samples were classified according to genetic ancestry using the Harmonized Ancestry and Race/Ethnicity (HARE) method[71]. GWAS analyses were performed on ancestry-stratified subsets in MVP using SAIGE[72] v1.1.6.2, adjusting for sex, age, mean-centered age-squared, and ten ancestry-specific principal components. To ensure accurate effect size estimation, Firth approximation was applied to single nucleotide polymorphisms (SNPs) with $P < 0.05$. Association scans were performed on well-imputed SNPs (INFO > 0.5) using an ancestry-specific MAF cutoff of ≥0.1% and a minimum minor allele count cutoff of 20.

## Local ancestry analysis at TCF4
Haplotype ancestry segments were inferred ("painted") in admixed populations using RFMix v2 with three rounds of expectation maximization and reference samples drawn from the 1000 Genomes Project and Human Genome Diversity Project (HGDP) reference panels[73]. Reference samples with ≥90% admixture in the population of interest were chosen. African-ancestry samples were painted using a two-way reference ($n = 631$ AFR, 695 EUR) and Hispanic/Latino-ancestry samples were painted using a three-way reference ($n = 631$ AFR, 695 EUR, 78 NAT). We then loaded the EUR ancestry dosage (0/1/2 corresponding to the number of EUR haplotypes) into VCFs. Finally, we tested the association of EUR ancestry dosage with FECD specifically at the *TCF4* locus (the locus most likely to demonstrate an admixture signal given the large effect size) separately in AFR and HIS cohorts, using SAIGE (v1.1.6.2), with the same model and covariates as used in the GWAS analyses.

## GWAS meta-analysis
We performed inverse variance-weighted fixed effects meta-analyses of GWAS summary statistics. First, we performed a EUR GWAS meta-analysis of MVP EUR and the ref. [14] discovery scan. (In Afshari et al., a GWAS was performed only on their discovery cohort of 1404 cases, and 2564 controls, whereas their replication analysis was performed on a selected set of variants significant in the discovery scan.) We then performed a multi-ancestry GWAS meta-analysis of MVP EUR, ref. [14], and MVP AFR. (MVP HIS was excluded from the multi-ancestry meta-analysis due to containing <100 cases.) Each set of summary statistics was converted into GWAS-VCFs using the +munge plug-in (https://github.com/freeseek/score) of bcftools[74] v1.16. The Afshari et al. summary statistics were lifted over to the GRCh38 genome build using the +liftover plug-in[75]. Finally, fixed-effect meta-analyses were performed using the +metal plug-in with an inverse-variance weighted scheme. For the multi-ancestry meta-analysis, a cohort-specific MAF ≥ 1% cutoff was applied. Manhattan plots were generated using the GWASLab Python package[76] as well as the Cmplot R package[77].

## Characterizing significant loci
We used the stepwise conditional and joint association analysis (COJO-slct) method implemented in GCTA[78] v1.94.1 to find conditionally independent genome-wide significant secondary signals at significant EUR meta-analysis loci. A linkage disequilibrium (LD) reference panel was constructed from 100,000 randomly selected MVP EUR subjects. The *TCF4* locus was excluded from COJO analysis due to the association of the untyped CTG18.1 repeat expansion. Variants for each independent genomic risk locus in the multi-ancestry meta-analysis were clumped and lead variants were identified using the Functional Mapping and Annotation of Genome-Wide Association Studies (FUMA) web server[79] (v1.4.2). The maximum $P$ value cutoff was set to 0.05, and a first LD threshold of $r^2 \geq 0.6$ and second threshold of $r^2 \geq 0.1$ were used to define loci and lead SNPs. The maximum distance between LD blocks to merge loci was 250 kb. Pleiotropy of significant loci with previous GWAS traits was identified using GWAS Catalog via FUMA, and Ldtrait[80], using a 250 kb range.

## LDSC
Non-partitioned liability score heritability for FECD and pairwise genetic correlations ($r_g$) between FECD and ocular traits were computed using LDSC[18] v1.0.1. Summary statistics for KC[30], CCT[47–49], and CRF[50], were obtained from GWAS Catalog; Pan-UK Biobank[52] CH summary statistics were obtained from https://pan.ukbb.broadinstitute.org; additional summary statistics for CRF and CH were provided by the authors[51]. Prior to computing $r_g$, all summary statistics were quality-controlled and alleles were harmonized to the reference genome using MungeSumstats[81] v1.7.8.

## SuSiE fine-mapping
Genome-wide significant loci in the EUR meta-analysis were fine-mapped using the sum of single effects (SuSiE)[22,23] v0.11.42. Pairwise SNP LD matrices were constructed from imputed dosages over the same sample set used in the MVP EUR GWAS ($N = 254,596$) using LDSTORE 2.0. Default options were used, including the maximum number of causal variants at a locus (10). The *TCF4* locus was excluded from this analysis due to the association of the untyped CTG18.1 repeat expansion.

## Associations of PGSs with FECD
Phenome-wide polygenic score files were obtained from European Molecular Biology Laboratory's European Bioinformatics Institute PGS Catalog[54]. All EUR subjects in MVP were scored across all available PGSs using the +score plugin (https://github.com/freeseek/score) of bcftools[74]. PGSs were then loaded into the dosage format field of VCFs readable by SAIGE for association testing. To determine pleiotropy of genetic predisposition to traits on FECD, logistic regression was used to examine associations of PGSs on MVP EUR FECD cases and controls using SAIGE[72] v1.1.6.2, adjusting for the same covariates as in GWAS (sex, age, mean-centered age-squared, and ten ancestry-specific principal components).

## PheWAS of index SNPs
We performed a PheWAS on each individual index SNP using summary statistics generated from the August 2022 beta release of the genome-wide PheWAS project in MVP[82]. Genotypes were imputed using the African Genome Resource and 1000 Genomes imputation panels. Phenotypes were derived from phecodes following standard definitions[56], a baseline survey distributed to all MVP enrollees, as well as EHR-based laboratory and vital signs measurements. A GWAS was performed on each phenotype in SAIGE using sex, age, age-squared, and 10 principal components as covariates.

## Colocalization
Genetic associations in MVP EUR participants at the *TCF4* locus (chr18:50,000,000 to 60,000,000 in hg38) for serum bicarbonate, chloride, and potassium were obtained using PLINK 2.0[83] alpha 4. Phenotypes were based on median clinical laboratory measurements recorded in the EHR. Traits were rank-based inverse normal transformed (RINT), and linear regression was performed using sex, age, mean-centered age-squared, and 10 principal components. Genotype QC was performed as in the FECD GWAS described above. Rank-based inverse normal transformed urinary

albumin-to-creatinine ratio (UACR) summary statistics[55] for EUR were obtained from GWAS Catalog (GCST008794) and lifted from hg19 to hg38. Effect comparison plots include only variants from chr18:54,500,000 to 56,500,000 with $P < 0.001$; effect correlation was measured using Pearson's $r$. Single causal variant colocalization was performed on summary statistics using the coloc.abf() function in coloc[58] v5.2.1. A posterior probability >0.9 for Hypothesis 4 (both traits are associated and share a single causal variant) was used as the criteria for colocalization.

## Structural analysis

Because no known crystal structures of human α5 LE22 and β1 LE6 exist, we modeled these using two protein structure prediction tools. SWISS-MODEL[60] was used to model the domains based on homology; the template with the highest Global Model Quality Estimation score was selected. AI-based AlphaFold 2[61] (AF2) was used to supplement SWISS-MODEL for the missing portions in the homology-based template (Supplementary Fig. 21). Structural differences between the SWISS-MODEL and the rat homolog, and those between SWISS-MODEL and AF2 predictions were both within the range of thermal fluctuations, lending confidence to the AF2 predictions. DUET was used to predict the change in protein stability due to the mutations[62].

## Statistics and reproducibility

For all analyses using FECD case-control status in MVP, the sample sizes are provided in Supplementary Data 1. For the index SNP PheWAS, the case and control sizes varied from phenotype to phenotype and are provided in Supplementary Data 8. GWAS replication was performed with an external cohort[14] of 1404 FECD cases and 2564 controls, which was combined with the MVP cohort in fixed-effect inverse variance-weighted meta-analyses. All statistical tests were two-tailed linear or logistic regressions, unless otherwise noted. Nominal significance was defined as $P < 0.05$. In hypothesis-free scans, we applied strict significance thresholds to account for multiple hypothesis testing. For GWAS analyses, the standard genome-wide significance threshold ($P < 5 \times 10^{-8}$) was used. In PheWAS analyses, we applied Bonferroni-corrected significance thresholds ($P < 0.05/560$ for the PGS scan and $P < 0.05/17,520$ for the index SNP PheWAS). All p-values are presented without adjustment for multiple hypotheses.

## Reporting summary

Further information on research design is available in the Nature Portfolio Reporting Summary linked to this article.

## Data availability

The full summary level association data from the meta-analysis and individual population association analyses in MVP are available via the dbGaP study accession number phs001672.

## Code availability

Software and analytical methods used in data analyses include SAIGE[72] v1.1.6.2 (https://github.com/weizhouUMICH/SAIGE) and PLINK2[83] alpha 4 (https://www.cog-genomics.org/plink/2.0) for genome-wide association analysis, munging and meta-analysis with bcftools[74] v1.16 (https://samtools.github.io/bcftools), conditional and joint association analysis using GCTA-COJO[78] v1.94.1 (https://yanglab.westlake.edu.cn/software/gcta/#COJO), heritability and genetic correlation analysis using LDSC[18] v1.0.1 (https://github.com/bulik/ldsc), fine-mapping with SuSiE[22,23] v0.11.42 (https://github.com/stephenslab/susieR), colocalization with the coloc R package[58] v5.2.1 (https://github.com/chr1swallace/coloc), protein modeling with AlphaFold 2[61] (https://github.com/google-deepmind/alphafold) and R v.4.2.2 for statistical analyses and plotting (https://www.r-project.org).

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

## Acknowledgements

We recognize the late Dr. Robert P. Igo (1965-2020) for his contributions in generating the summary statistics from ref. [14] used in this work. This research is based on data from the Million Veteran Program, Office of Research and Development, Veterans Health Administration, and was supported by award I01 BX003364. This work was also supported by the Cleveland Institute for Computational Biology, NIH Core Grants (P30 EY025585, P30 EY011373), the Clinical and Translational Science Collaborative of Cleveland (UL1TR002548) from the National Center for Advancing Translational Sciences (NCATS) component of the NIH and NIH Roadmap for Medical Research, the VA Research Career Scientist award (IK6 BX005233; N.S.P.), unrestricted grants from Research to Prevent Blindness to Case Western Reserve University, Cleveland Clinic Lerner College of Medicine of Case Western Reserve University, the Harper-Inglis Memorial for Eye Research, The Peierls Foundation, and That Man May See. We thank the Veteran participants in MVP and MVP staff. This publication does not represent the views of the Department of Veteran Affairs or the United States Government. We acknowledge the VA Million Veteran Program (MVP) and the VA-DOE genome-wide PheWAS core analytic team for generating the corresponding PheWAS summary statistics that were used in this manuscript. Support for title page creation and format was provided by AuthorArranger, a tool developed at the National Cancer Institute.

## Author contributions

Drafted the manuscript: B.R.G., M.F., N.S.P., S.K.I. Analyzed the data: B.R.G., M.F., N.D., K.M., G.G. Acquired the data: B.R.G., C.L.N., C.W.H., P.G.H., H.C., N.A.A., Y.-J. L., J.M.G., VA MVP, S.P., N.S.P., S.K.I. Critically revised the manuscript for important intellectual content: B.R.G., M.F., C.L.N., C.W.H., N.D., K.M., G.G., P.G.H., H.C., N.A.A., Y.-J.L., J.M.G., A.M.H., W.-C.W., P.B.G., S.P., J.H.L., N.S.P., S.K.I.

## Competing interests

Hélène Choquet is an Editorial Board Member for *Communications Biology*, but was not involved in the editorial review, nor the decision to publish this article. All other authors have declared no competing interests.

## Additional information

[1]Center for Data and Computational Sciences (C-DACS), VA Boston Healthcare System, Boston, MA, USA. [2]Booz Allen Hamilton, McLean, VA, USA. [3]Eye Clinic, VA Northeast Ohio Healthcare System, Cleveland, OH, USA. [4]Center of Innovation in Long Term Services and Supports, Providence VA Medical Center, Providence, RI, USA. [5]Program in Medical and Population Genetics, Broad Institute of MIT and Harvard, Cambridge, MA, USA. [6]Stanley Center, Broad Institute of MIT and Harvard, Cambridge, MA, USA. [7]Department of Genetics, Harvard Medical School, Boston, MA, USA. [8]Department of Ophthalmology, King's College London, London, UK. [9]Department of Twins Research and Genetic Epidemiology, King's College London, London, UK. [10]UCL Great Ormond Street Hospital Institute of Child Health, King's College London, London, UK. [11]Division of Research, Kaiser Permanente Northern California (KPNC), Oakland, CA, USA. [12]Shiley Eye Institute, Viterbi Family Department of Ophthalmology, University of California, San Diego, La Jolla, CA, USA. [13]Department of Biostatistics and Bioinformatics, Duke University School of Medicine, Durham, NC, USA. [14]Massachusetts Veterans Epidemiology Research and Information Center (MAVERIC), VA Boston Healthcare System, Boston, MA, USA. [15]Division of Aging, Department of Medicine, Brigham and Women's Hospital, Harvard Medical School, Boston, MA, USA. [16]Division of Nephrology and Hypertension, Department of Medicine, Vanderbilt University Medical Center, Nashville, TN, USA. [17]Vanderbilt Center for Kidney Disease, Vanderbilt University Medical Center, Nashville, TN, USA. [18]VA Tennessee Valley Healthcare System, Nashville, TN, USA. [19]Cardiology Section, Medical Service, Providence VA Medical Center, Providence, RI, USA. [20]Ophthalmology Section, Providence VA Medical Center, Providence, RI, USA. [21]Division of Ophthalmology, Alpert Medical School, Brown University, Providence, RI, USA. [22]Department of Ophthalmology and Visual Sciences, Case Western Reserve University, Cleveland, OH, USA. [23]Research Service, VA Northeast Ohio Healthcare System, Cleveland, OH, USA. [24]Cole Eye Institute, Cleveland Clinic Foundation, Cleveland, OH, USA. [25]Department of Ophthalmology, Cleveland Clinic Lerner College of Medicine of Case Western Reserve University, Cleveland, OH, USA. [26]Cleveland Institute for Computational Biology, Case Western Reserve University, Cleveland, OH, USA. [27]Department of Population and Quantitative Health Sciences, Case Western Reserve University School of Medicine, Cleveland, OH, USA. [31]These authors contributed equally: Bryan R. Gorman, Michael Francis. [32]These authors jointly supervised this work: Neal S. Peachey, Sudha K. Iyengar. ✉e-mail: neal.peachey@va.gov; ski@case.edu

## VA Million Veteran Program

**J. Michael Gaziano[14,15], Philip S. Tsao[28,29,30] & Saiju Pyarajan ⓘ[1]**

[28]VA Palo Alto Epidemiology Research and Information Center for Genomics, VA Palo Alto Health Care System, Palo Alto, CA 94304, USA. [29]Department of Medicine, Stanford University School of Medicine, Stanford, CA 94305, USA. [30]Stanford Cardiovascular Institute, Stanford University School of Medicine, Stanford, CA 94305, USA.

