## [Peer Review File · Communications Biology]

Reviewers' comments:

Reviewer #1 (Remarks to the Author):

Multi-ancestry GWAS of Fuchs corneal dystrophy highlights roles of laminins, collagen, and endothelial cell regulation

The authors have presented one of the largest GWA study on FECD. The study cohort includes 2646 cases from Million Veteran Program including EUR, AFR and HIS ancestry and 357,598 matched controls and a replication and meta-analysis from a previous study with a total of 3970 cases and 333,794 controls. They have identified 8 novel loci and confirmed association of previously identified 4 loci with the TCF4 rs11659764 as the lead variant across all ancestries. In the MVP cohort 7 loci, 4 previously identified loci and three novel loci were identified. The three novel loci at SSBP3, THSD7A, PIDD1 had reached nominal significance in the previous study too. Further multi-ancestry meta-analysis with the earlier cohort (Afsari et al) has identified 8 novel loci LAMA5, LAMB1, COL18A1, SSBP3, THSD7A, RORA, PIDD1, and HS3ST3B1 which includes the three novel loci in the current discovery cohort. The novel loci/genes and their potential role in corneal endothelial cell functioning with reference to available literature has been detailed. The possible structural changes of the protein due to the two missense variants in LAMA5 and LAMB1 have been presented. Involvement of some of the loci in other ocular traits including corneal traits have been presented. The effect size and direction of the FECD lead variants have been compared to other corneal traits as well. Polygenic risk scores catalogue based PGS with FECD status have identified several significant renal PGSs further to corneal traits. The PheWAS has indicated involvement of TCF4 loci in renal function and is substantiated by laboratory measurements.

The introduction is brief and comprehensive, the methodology well detailed and the results elaborately described.

The authors have listed the limitations of the study as well, the major one being the phenotype confirmation of the cases by EMR and not be slit lamp imaging.

The study presents GWAS on FECD with the largest cohort till date. The identification 8 novel loci, confirmation of previously identified 4 loci, totaling to 12 loci will further help to understand the biology of the disease and is a significant contribution to the studies on FECD.

Comments

1. The cohort size of the study, both the current discovery cohort and the meta-analysis cohort needs to be mentioned in the abstract.
2. The novel loci too may be listed in the abstract.
3. Introduction is crisp and comprehensive, however, very brief details on of early -onset FECD, inheritance and gene(s) involved would complete the literature available on the disease.
4. There are five novel loci (LAMA5, LAMB1, COL18A1, RORA, HS3ST3B1) identified in the meta-analysis, were these loci looked again in the discovery cohort for GWS? If so, was GWS reached? If there was nominal significance, then probably these loci do not necessarily indicate a bias towards male specific genetic factors as the previous cohort has higher percentage of females which is case with FECD in general.
5. Were the novel loci specifically again analyzed for in the previous cohort alone (Afsari et al)?
6. Was polygenic risk score done/attempted with the genotype and phenotype data available with the meta-analysis cohort?

Reviewer #2 (Remarks to the Author):

The authors present GWAS analyses of Fuchs endothelial corneal dystrophy (FECD) that deliver several novel important findings- those are clearly indicated in the abstract: 1) a prominent biologically and genetically supported role for laminin 511 2) a risk conferred across ancestries by the TCF4 locus driven by European-ancestry haplotypes and 3) a comprehensive survey of pleiotropic effects of FECD associations, in particular that of the TCF4 locus on ion transport in the kidney. They thus provide a landmark paper for FECD genetics capitalizing on sample size and ancestry diversity gained from analysing the Million Veteran Program (MVP) emerging mega-Biobank. The manuscript is overall well-written and standard methods in the field applied.

My main comment is on the presentation of some of the FECD GWAS results. The authors choose to make emphasis on the multi-ancestry nature of the sample analysed, sometimes - title, Manhattan plot for multi-ancestry GWAS meta-analysis of GWAS as a main figure- a bit disingenuously on the impact that made. We all appreciate the need to extend analysis to diverse ancestries, however the numbers of cases from non European ancestries are still low in this study and thus contributed little to the discovery of novel FECD loci reported from the meta-analysis combining MVP results to those of previously reported GWAS studies (European ancestry). For the meta-analysis results presentation, a narrative with the results obtained with the largest represented ancestry group- European- fully displayed before the "multi ancestry GWAS" would help evaluate genetic findings- Manhattan plot Figure 2 could be replaced by Miami plot with European ancestry and "multi ancestry" mirroring each other. There is some confusion as well introduced by the dataset referred to as "Afshari et al 2017" representing sometimes the meta-analysed European sample analysed at the time (2,075 cases and 3,342 controls.) – lines 230 to 234- and other times just the discovery (1,404 cases, 2,564 controls) as presented Figure 1)-line 236, figure 1 etc . This means that the 2.8 times increase sample size is not quite right line 240 and replication confusing as replication as presented figure 1 expected. The "replicated" -line 242- for the multi-ancestry meta-analysis results is not correct to use here: the original study it replicates (or rather part of it) being included in the studies meta-analysed.

Minor points are listed below:

Line 63-64: Lack of donor cornea and rejection rate said to demonstrate need for early detection of FECD susceptibility- How would it help?

Line 103: case algorithm principle to be briefly detailed here

Line 118: MAF cut off 0.1% seems low for small sample (ancestry) –corresponding minimum allele count in cases ?

Line 129 Characterizing significant loci paragraph- clarify if this all done on the European meta-analysis results

Line 142- Non-partitioned liability score heritability- what would the partitioning be on?

Line 175: age not mean-centered age squared used interchangeably use 1- 20 pcs not 10pcs here-Is there some justification for variable choice across manuscript?

Line 215 : reference or description in methods needs to be provided for local ancestry mapping models- would method be applicable in the European samples ?

Line 256: "Two AFR meta-analysis SNPs" to rephrase

COLXVIII: given other hits as laminins worth mentioning that COLXVIII " isoforms share an N-terminal laminin-G-like/thrombospondin-1 sequence whose specific functions still remain unconfirmed."

Following ref 51

Line 280 LAMB1 no pleiotropic effect from GWAS catalog but later authors cite association with heart rate in UK Biobank - Consider expanding survey to GWAS results repositories that include UKB

Line 308: Not clear how candidate genes were linked to SNP in this gene rich region

Line 347 may add the nominal association with FECD reported in previous look up of CCT associated

variants – at then called AVGR8 locus(Igo et al,2012 PLOs one)

Line 357 potential bidirectional causality –unclear

Line 368 consistent with previous report: consistent with expectation as reference cited not a report on genetic effect directions

Line 394 negative association with glaucoma- what about IOP? no association?

Figure 1: Horizontal arrow to variant effects comparisons, PheWAS etc should start from European ancestry cartouche rather than multi-ancestry

Supplemental figures legend generally too succinct

FigS2 what was the LD reference used? would it not make more sense to plots EUROPEAN only for display in locuszoom plots?

Point-by-point response to reviewers:

Multi-ancestry GWAS of Fuchs corneal dystrophy highlights roles of laminins, collagen, and endothelial cell regulation

COMMSBIO-23-1302-T

Referee expertise:

Referee #1: Retinal genetics, GWAS

Referee #2: Genomics of retinal and metabolic disorders

Reviewers' remarks are presented in italics

>Our response in non-italics and colored blue.

>>> We thank the reviewers for their highly supportive comments and suggestions, which have been helpful in improving the quality of our work.

Reviewer #1 (Remarks to the Author):

The authors have presented one of the largest GWA study on FECD. The study cohort includes 2646 cases from Million Veteran Program including EUR, AFR and HIS ancestry and 357,598 matched controls and a replication and meta-analysis from a previous study with a total of 3970 cases and 333,794 controls. They have identified 8 novel loci and confirmed association of previously identified 4 loci with the TCF4 rs11659764 as the lead variant across all ancestries. In the MVP cohort 7 loci, 4 previously identified loci and three novel loci were identified. The three novel loci at SSBP3, THSD7A, PIDD1 had reached nominal significance in the previous study too. Further multi-ancestry meta-analysis with the earlier cohort (Afsari et al) has identified 8 novel loci LAMA5, LAMB1, COL18A1, SSBP3, THSD7A, RORA, PIDD1, and HS3ST3B1 which includes the three novel loci in the current discovery cohort. The novel loci/genes and their potential role in corneal endothelial cell functioning with reference to available literature has been detailed. The possible structural changes of the protein due to the two missense variants in LAMA5 and LAMB1 have been presented. Involvement of some of the loci in other ocular traits including corneal traits have been presented. The effect size and direction of the FECD lead variants have been compared to other corneal traits as well. Polygenic risk scores catalogue based PGS with FECD status have identified several significant renal PGSs further to corneal traits. The PheWAS has indicated involvement of TCF4 loci in renal function and is substantiated by laboratory measurements.

The introduction is brief and comprehensive, the methodology well detailed and the results elaborately described.

The authors have listed the limitations of the study as well, the major one being the phenotype confirmation of the cases by EMR and not by slit lamp imaging.

The study presents GWAS on FECD with the largest cohort till date. The identification 8 novel loci, confirmation of previously identified 4 loci, totaling to 12 loci will further help to understand the biology of the disease and is a significant contribution to the studies on FECD.

We thank the reviewer for their positive remarks.

Comments

1. The cohort size of the study, both the current discovery cohort and the meta-analysis cohort needs to be mentioned in the abstract.

We thank the reviewer for this comment. We have added the sample size of the multi-ancestry meta-analysis to the abstract (Line 48). Because there were three ancestry-stratified discovery cohorts in this study, which were not all directly meta-analyzed into a single cohort, we felt adding the discovery sample sizes could potentially introduce confusion.

2. The novel loci too may be listed in the abstract.

We thank the reviewer for this comment. We have added the list of novel loci to the abstract (Line 49).

3. Introduction is crisp and comprehensive, however, very brief details on of early -onset FECD, inheritance and gene(s) involved would complete the literature available on the disease.

We thank the reviewer for this comment. We have added this sentence to provide brief details on early-onset FECD (Line 85):

“Mutations in *COL8A2* cause the rarer early-onset form of FECD, which has a similar disease progression to late-onset FECD, but is characterized by an abnormal distribution of collagen VIII¹².”

4. There are five novel loci (LAMA5, LAMB1, COL18A1, RORA, HS3ST3B1) identified in the meta-analysis, were these loci looked again in the discovery cohort for GWS? If so, was GWS reached? If there was nominal significance, then probably these loci do not necessarily indicate a bias towards male specific genetic factors as the previous cohort has higher percentage of females which is case with FECD in general.

We thank the reviewer for this comment. We have generated a new table (Supplementary Table 3) to facilitate comparison of summary statistics across meta-analyses and individual cohorts for the twelve genome-wide significant index variants. Additionally we have added this sentence to the discussion (Line 441):

“However, because our novel index variants were all at least nominally significant in Afshari et al.¹⁴ (68% female FECD cases), with consistent effect estimates (heterogeneity $P > 0.05$; Supplementary Fig. 3), our results may indeed be generalizable to both male and female.”

5. Were the novel loci specifically again analyzed for in the previous cohort alone (Afsari et al)?

We thank the reviewer for this question. This information is provided in Supplementary Fig. 3, and was addressed in Line 160:

“As expected, the largest OR was observed at rs11659764 in TCF4 (OR=7.15 [6.60, 7.74]; Supplementary Fig. 3). Effect sizes at index SNPs were consistent across the MVP EUR and Afshari¹⁴ cohorts, and all meta-analysis index SNPs were at least nominally significant ($P < 0.05$) in the prior GWAS, further validating our phenotyping approach.”

Additionally, the newly added Supplementary Table 3 mentioned in our previous comment response can facilitate easier comparisons of associations across cohorts.

6. Was polygenic risk score done/attempted with the genotype and phenotype data available with the meta-analysis cohort?

We thank the reviewer for this suggestion. Evaluation of polygenic risk prediction models for FECD, requiring a genotyped and phenotyped independent cohort, is part of our planned future work.

Reviewer #2 (Remarks to the Author):

The authors present GWAS analyses of Fuchs endothelial corneal dystrophy (FECD) that deliver several novel important findings- those are clearly indicated in the abstract: 1) a prominent biologically and genetically supported role for laminin 511 2) a risk conferred across ancestries by the TCF4 locus driven by European-ancestry haplotypes and 3) a comprehensive survey of pleiotropic effects of FECD associations, in particular that of the TCF4 locus on ion transport in the kidney. They thus provide a landmark paper for FECD genetics capitalizing on sample size and ancestry diversity gained from analysing the Million Veteran Program (MVP) emerging mega-Biobank. The manuscript is overall well-written and standard methods in the field applied.

We thank the reviewer for their overall positive review of our manuscript. In addition to addressing the reviewer's points below, we have edited parts of the discussion in order to better emphasize the themes highlighted by the reviewer.

My main comment is on the presentation of some of the FECD GWAS results. The authors choose to make emphasis on the multi-ancestry nature of the sample analysed, sometimes - title, Manhattan plot for multi-ancestry GWAS meta-analysis of GWAS as a main figure- a bit disingenuously on the impact that made. We all appreciate the need to extend analysis to diverse ancestries, however the numbers of cases from non European ancestries are still low in this study and thus contributed little to the discovery of novel FECD loci reported from the meta-analysis combining MVP results to those of previously reported GWAS studies (European ancestry). For the meta-analysis results presentation, a narrative with the results obtained with the largest represented ancestry group- European- fully displayed before the "multi ancestry GWAS" would help evaluate genetic findings- Manhattan plot Figure 2 could be replaced by Miami plot with European ancestry and "multi ancestry" mirroring each other.

We thank the reviewer for this comment. Our analysis provides important information about FECD in the context of non-European populations, and justifies our choice of characterizing our study as multi-ancestry. We performed the first GWAS of FECD in African and Hispanic groups, and identified genome-wide significance of the *TCF4* locus in these groups for the first time. Although overall risk was lower in our African and Hispanic cohorts, the sample sizes of these were larger than the sample size of the original GWAS cohort that discovered *TCF4*. We also performed a local ancestry analysis at *TCF4*, which showed a novel association with European-ancestry haplotypes in admixed African ancestry. Importantly, our results showed strong consistency between European and African groups at FECD risk loci; at five of the six index variants meeting a 1% frequency threshold in the African cohort, we obtained a more significant effect

estimate after meta-analysis. Thus, we highlighted the multi-ancestry GWAS result because it can be more broadly interpreted across populations. We have also facilitated easier comparison between the European-only meta-analysis, the multi-ancestry meta-analysis, and individual cohorts, with the addition of a new Supplementary Table.

There is some confusion as well introduced by the dataset referred to as “Afshari et al 2017” representing sometimes the meta-analysed European sample analysed at the time (2,075 cases and 3,342 controls.) – lines 230 to 234- and other times just the discovery (1,404 cases, 2,564 controls) as presented Figure 1)-line 236, figure 1 etc . This means that the 2.8 times increase sample size is not quite right line 240 and replication confusing as replication as presented figure 1 expected.

We thank the reviewer for pointing out this potential source of confusion. The discovery GWAS in Afshari et al. was performed using 1,404 cases and 2,564 controls. The replication phase of that study performed PCR genotyping of a selected set of markers that were identified in the discovery scan. Therefore, our characterization of our GWAS study as “~2.8 times the case sample size of the previous largest FECD GWAS cohort” was accurate, as the replication cohort from Afshari was not analyzed in a GWAS.

We have added the following sentence to the methods to clarify this point (Line 514):

“(In Afshari et al., GWAS was only performed on their discovery cohort of 1,404 cases, and 2,564 controls, whereas their replication and meta-analysis were performed using a selected set of significant variants.)”

The “replicated” -line 242- for the multi-ancestry meta-analysis results is not correct to use here: the original study it replicates (or rather part of it) being included in the studies meta-analysed.

We thank the reviewer for this comment; we agree a meta-analysis does not inherently replicate its constituent studies. The sentence in question has been updated to read as follows (Line 152):

“In the multi-ancestry meta-analysis, the four previously reported loci remained significant, and we confirmed the eight novel FECD loci emerging at GWS from the EUR meta-analysis: *LAMA5*, *LAMB1*, *COL18A1*, *SSBP3*, *THSD7A*, *RORA*, *PIDD1*, and *HS3ST3B1* (Table 2; Supplementary Table 3; Fig. 2; Supplementary Fig. 2).”

Minor points are listed below:

- *Line 63-64: Lack of donor cornea and rejection rate said to demonstrate need for early detection of FECD susceptibility- How would it help?*

We thank the reviewer for this comment. We have modified this sentence in the introduction to clarify this point. It now reads (Line 65):

“As surgical and pharmaceutical therapies are developed, genetically informed early diagnosis of FECD will be critical for directing treatment and preventing irreversible damage.”

- *Line 103: case algorithm principle to be briefly detailed here*

We thank the reviewer for this comment. We have updated the Phenotyping section of Methods to read as follows (Line 478):

“*Phenotyping.* We used a rules-based algorithm⁹ based on structured electronic health record (EHR) data (International Classification of Diseases Clinical Modification (ICD-CM) and Current Procedural Terminology (CPT)) codes, the accuracy of which was confirmed at three VA Medical Center Eye Clinics⁹. Cases were identified based on the presence of FECD codes (371.57 for ICD-9-CM; H18.51 for ICD-10-CM) on two separate visits and the absence of ICD-9-CM or ICD-10-CM codes for confounding corneal conditions or complicated intraocular surgeries. Controls without FECD were identified as having undergone at least one eye exam, with no codes for FECD, confounding corneal conditions, or complicated intraocular surgeries. We applied this algorithm to conduct GWAS and to analyze associated EHR data.”

- *Line 118: MAF cut off 0.1% seems low for small sample (ancestry) –corresponding minimum allele count in cases ?*

We thank the reviewer for identifying this oversight in reporting the GWAS genotype quality control methods. The sentence in question has been updated to include the minimum minor allele count (mac) cutoff as follows (Line 499):

“Association scans were performed on well-imputed SNPs (INFO>0.5) using an ancestry-specific minor allele frequency (MAF) cutoff of $\geq 0.1\%$ and a minimum minor allele count (MAC) cutoff of 20.”

- *Line 129 Characterizing significant loci paragraph- clarify if this all done on the European meta-analysis results*

We thank the reviewer for pointing this out. COJO-slct was performed on the EUR meta-analysis summary statistics, as this method depends on a higher degree of ancestral similarity between the GWAS sample and the LD reference panel. Meanwhile, as indicated, FUMA genomic risk locus identification was performed using the multi-ancestry meta-analysis summary statistics. We have updated the relevant sentence (Line 526):

“We used the stepwise conditional and joint association analysis (COJO-slct) method implemented in GCTA⁷⁶ v1.94.1 to find conditionally independent genome-wide significant secondary signals at significant EUR meta-analysis loci.”

- *Line 142- Non-partitioned liability score heritability- what would the partitioning be on?*

We used the term “non-partitioned heritability” to distinguish our estimate from partitioned liability, which refers to calculating heritability while accounting for the functional categories of the SNPs involved in the calculation. (Finucane, H., Bulik-Sullivan, B., Gusev, A. *et al.* Partitioning heritability by functional annotation using genome-wide association summary statistics. *Nat Genet* 47, 1228–1235 (2015). <https://doi.org/10.1038/ng.3404>; <https://github.com/bulik/ldsc/wiki/Partitioned-Heritability>)

However, as non-partitioned heritability would be assumed by most readers unless otherwise specified, to avoid confusion we have simply removed the “non-partitioned” descriptor from that sentence (Line 170).

- *Line 175: age not mean-centered age squared used interchangeably use 1- 20 pcs not 10pcs here-Is there some justification for variable choice across manuscript?*

We thank the reviewer for pointing out this unintentional discrepancy. We have previously found that the population structure in our cohort is sufficiently captured with 10 PCs calculated within the ancestry being analyzed. Therefore, we have re-run the bicarbonate, chloride, and potassium analyses at *TCF4* using the same covariates as the FECD GWAS, and updated Supplementary Figure 7 and Supplementary Table 8 with the new results (which are near-identical with the existing results and thus do not require re-interpretation). Additionally, we have updated this sentence of the methods (Line 572):

“Genetic associations in MVP EUR participants at the *TCF4* locus (chr18:50,000,000 to 60,000,000 in hg38) for serum bicarbonate, chloride, and potassium were obtained using PLINK 2.0 alpha 4⁸¹. Phenotypes were based on median clinical laboratory measurements recorded in the EHR. Traits were rank-based inverse normal transformed (RINT), and linear regression was performed using sex, age, mean-centered age-squared, and 10 principal components.”

- *Line 215 : reference or description in methods needs to be provided for local ancestry mapping models- would method be applicable in the European samples ?*

We thank the reviewer for identifying this oversight. Methods have been added under the heading “Local ancestry analysis at *TCF4*” (Line 500, see below). For this paper, we have focused on enrichment of haplotypes at *TCF4* classified on the basis of continental ancestry. A similar analysis applied to sub-continental ancestry could be an interesting future direction, but would be challenging due to smaller differences in allele frequencies across sub-continental populations.

Local ancestry analysis at TCF4. Haplotype ancestry segments were inferred (“painted”) in admixed populations using RFMix v2 with three rounds of expectation maximization and reference samples drawn from the 1000 Genomes Project and Human Genome Diversity Project (HGDP) reference panels⁷². Reference samples with ≥90% admixture in the population of interest were chosen. African-ancestry samples were painted using a two-way reference (n=631 AFR, 695 EUR) and Hispanic/Latino-ancestry samples were painted using a three-way reference (n=631 AFR, 695 EUR, 78 NAT). We then loaded the EUR ancestry dosage (0/1/2 corresponding to the number of EUR haplotypes) into VCFs. Finally, we tested the association of EUR ancestry dosage with FECD specifically at the *TCF4* locus (the locus most likely to demonstrate an admixture signal given the large effect size) separately in AFR and HIS cohorts, using SAIGE (v1.1.6.2), with the same model and covariates as used in the GWAS analyses.

- *Line 256: “Two AFR meta-analysis SNPs” to rephrase*

We thank the reviewer for identifying this awkward phrasing. We have removed “meta-analysis” from this clause (Line 167):

“Two AFR SNPs, rs1138714 in *PIDD1* and rs114065856 in *COL18A1*, had consistent direction with EUR cohorts but were not significant.”

- *COLXVIII: given other hits as laminins worth mentioning that COLXVIII “ isoforms share an N-terminal laminin-G-like/thrombospondin-1 sequence whose specific functions still remain unconfirmed.” Following ref 51*

We thank the reviewer for this observation. We have added the following sentence to the discussion of COL18A1 (Line 404):

“Type XVIII collagen contains a laminin-G-like/thrombospondin-1 (LAM-G/TSP-1) homology region and thus exhibits structural similarity to laminins and thrombospondins such as THSD7A³².”

- *Line 280 LAMB1 no pleiotropic effect from GWAS catalog but later authors cite association with heart rate in UK Biobank - Consider expanding survey to GWAS results repositories that include UKB*

We thank the reviewer for this suggestion. On line 280, we have identified a missing word as this sentence was meant to refer strictly to other ocular traits. Therefore we have updated this to read (Line 190):

“Interestingly, the LAMB1 locus has no pleiotropy with other ocular traits reported in the GWAS Catalog (Supplementary Table 4).”

Regarding the reference to the PheWAS association with heart rate, our intention was to point out an interesting pleiotropic association of *LAMB1* that was discovered in our MVP-based PheWAS which was replicated in UK Biobank. After further review, we determined that this was the only instance in our analysis where the GWAS Catalog did not contain a significant association that was otherwise reported in a UK Biobank PheWAS.

- *Line 308: Not clear how candidate genes were linked to SNP in this gene rich region*

We thank the reviewer for this comment. We have expanded our fine-mapping analysis to include this locus, and revised the text accordingly (Lines 219-233). The index SNP is in a 3' UTR of *PNPLA2*. Given that the nearest gene to the sentinel SNP in a locus is the causal gene about 70% of the time (<https://www.nature.com/articles/s41586-021-04103-z>), we considered *PNPLA2* a plausible candidate gene. Similarly, many of the SNPs in the 95% credible set, including those with the highest CADD scores, were either within *PIDD1* or are closer to *PIDD1* than to other genes; we thus prioritized *PIDD1*—in addition to its function as a regulator of apoptosis, which is an important element of FECD etiology. Finally, the function of *CD151* (basement membrane assembly) is perhaps the

most relevant to other FECD genes, and also has an eQTL association with the index SNP. On balance, we felt the fine-mapping results slightly favored *PIDD1*, so we have updated this paragraph accordingly, and changed other places such as Fig. 2 and Tables 2 and 3 to name this locus only by *PIDD1* instead of “*CD151/PIDD1*.”

- *Line 347 may add the nominal association with FECD reported in previous look up of CCT associated variants – at then called AVGR8 locus(Igo et al,2012 PLOs one)*

We thank the reviewer for this comment. With regard to our chr17 locus (*HS3ST3B1*) discussed on line 347, there is no corresponding association in this paper mentioned (Differing Roles for TCF4 and COL8A2 in Central Corneal Thickness and Fuchs Endothelial Corneal Dystrophy. Igo RP Jr, Kopplin LJ, Joseph P, Truitt B, Fondran J, et al. (2012) Differing Roles for TCF4 and COL8A2 in Central Corneal Thickness and Fuchs Endothelial Corneal Dystrophy. PLOS ONE 7(10): e46742. <https://doi.org/10.1371/journal.pone.004674>). *AVGR8* is on chr13, and additionally none of the variants described in this paper are near our genome-wide significant hits, with the exception of *TCF4*.

- *Line 357 potential bidirectional causality –unclear*

We thank the reviewer for pointing out this potentially confusing language. We have removed the clause in question so that the sentence now reads (Line 271):

“Thus, our results support a complex relationship between CEC density and FECD.”

- *Line 368 consistent with previous report: consistent with expectation as reference cited not a report on genetic effect directions*

We thank the reviewer for this comment. To clarify, we have split this sentence into two and noted that the previous reports did not refer to genetic effects (Line 280):

“At the nominally significant variants for each respective trait, all KC and CCT variant effects were in the same direction as FECD, while all CRF variants, and all variants but one in CH (*SSBP3*) were associated with effects in the opposite direction (Fig. 3; Supplementary Table 5). The relationship of genetic effects of CRF and CH with those of FECD were directionally consistent with previous observational reports⁶⁶”

- *Line 394 negative association with glaucoma- what about IOP? no association?*

We thank the reviewer for this question. IOP is not currently available in the electronic health record data provided to us in MVP.

- *Figure 1: Horizontal arrow to variant effects comparisons, PheWAS etc should start from European ancestry cartouche rather than multi-ancestry*

We thank the reviewer for this comment. Our rationale for selecting variants as input for the follow-up analyses indicated in Figure 1 was selection of the index variants from the final multi-ancestry meta-analysis. Therefore, the arrow as drawn correctly represents the design of our analysis.

- *Supplemental figures legend generally too succinct*

We thank the reviewer for this comment. We have updated the Supplemental Figure legends to include more details. A file with tracked changes to these legends has also been provided.

- *FigS2 what was the LD reference used? would it not make more sense to plots EUROPEAN only for display in locuszoom plots?*

We thank the reviewer for this suggestion. We used the European LD reference panel for all plots, this has been indicated in the revised figure caption. We decided the EUR panel was a closer match to our mixed EUR and AFR meta-analysis results than "ALL." However, we chose to plot the multi-ancestry meta-analysis GWAS results in this supplementary figure because this represented our primary set of results, as described in the comment responses above.

REVIEWERS' COMMENTS:

Reviewer #1 (Remarks to the Author):

The authors have addressed the comments and have modified the manuscript along with tables, figures and legends.

Reviewer #2 (Remarks to the Author):

Revised manuscript: Apart from phrasing below the authors addressed all points raised satisfactorily and I have no further comments. The supplementary table 3 is a worthy addition for clarification of results obtained and for future studies.

"In the multi-ancestry meta-analysis, the four previously reported loci remained significant, and we confirmed the eight novel FECD loci emerging at GWS from the EUR meta-analysis: LAMA5, LAMB1, COL18A1, SSBP3, THSD7A, RORA, PIDD1, and HS3ST3B1 (Table 2; Supplementary Table 3; Fig. 2; Supplementary Fig. 2)."

At face value, the fact that the four loci remained significant by adding a study of small size to a larger set does not give the reader any information. Additionally, it is not clear what is meant by confirmation of the 8 novel loci since the multi-ancestry meta-analysis included only 4 of the 8 GWS variants from the EUR meta-analysis.